# Effectiveness of online practical education on vaccination training in the students of bachelor programs during the Covid-19 pandemic

Samane Shirahmadi[1]☯, Seyed Mohamad Mehdi Hazavehei[2], Hamid Abbasi[3], Marzie Otogara[4], Tahere Etesamifard[5], Ghodratolah Roshanaei[6], Neda Dadaei[7], Malihe Taheri [5]☯*

1 Department of Community Oral Health and Dental Research Centers, School of Dentistry, Hamadan University of Medical Sciences, Hamadan, Iran, 2 Department of Public Health, Hamadan University of Medical Sciences, Hamadan, Iran, 3 Department of Public Health, School of Public Health, Neyshabur University of Medical Sciences, Neyshabur, Iran, 4 Department of Clinical Research Development, Fatemieh Hospital, Hamadan University of Medical Sciences, Hamadan, Iran, 5 Department of Public Health, School of Public Health, Hamadan University of Medical Sciences, Hamadan, Iran, 6 Department of Biostatistics, School of Public Health, Modeling of Noncommunicable Diseases Research Center, Hamadan University of Medical Sciences, Hamadan, Iran, 7 Departments of Oral Health, Vice Chancellor for Health, Hamadan University of Medical Sciences, Hamadan, Iran

☯ These authors contributed equally to this work.
* ma.taheri@umsha.ac.ir

**Data Availability Statement:** All relevant data are within the paper and its Supporting Information files.

## Abstract

### Background

The importance of immunization and the necessity of achieving the goals of the immunization expansion plan and the critical role of undergraduate public health students in attaining these goals in the Covid-19 pandemic is evident. The present study aimed at investigating the effectiveness of using online educational videos on practical learning of vaccination in the apprenticeship stage during covid-19 pandemic: a randomized controlled trial.

### Material and methods

This experimental study was conducted on 120 students (60 interventions and 60 control groups) at Hamadan University of Medical Sciences during 2019–2020. The intervention included training vaccination skills through educational videos based on self-efficacy theory, which was conducted for two weeks each week in two sessions of two hours for the intervention group using an educational video. A researcher-made questionnaire and a performance checklist were used to collect data. Data were analyzed using SPSS-16 software. Paired t-test, independent t-test, and Chi-square.

### Results

The mean age of the subjects was 22.41 years, and most of the participants were female students (80%). There were statistically significant differences between the intervention and

**Funding:** The current study was supported by Vice Chancellor for Research and Technology, Hamadan University of Medical Sciences [grant numbers: 9407213959- Recipient: Seyed Mohamad Mehdi Hazavehei]. The funding body had no role in study design, data collection and analysis, decision to publish, or preparation of the manuscript.

**Competing interests:** The authors have declared that no competing interests exist.

control groups regarding knowledge (19.17±0.92 vs. 16.03±3.00; P<0.001), self-efficacy (40.84±3.71 vs 33.45±4.83; P = 0.01), attitude (22.56±2.95vs 20.28±3.25; P = 0.01) and performance (27.92±6.00 vs 22.38±5.40; P = 0.01) after the intervention.

## Conclusion

According to the findings of this study, the use of educational videos for undergraduate students of public health during the apprenticeship period has a positive effect on the practical learning of vaccination. However, it seems that in non-critical times, online education along with face-to-face education will be more effective for practical training.

## Introduction

COVID-19 caused by a novel coronavirus (2019-nCoV) was declared a pandemic disease by the World Health Organization (WHO) in March 2020 [1]. Presently, there are more than 79 million infected people and 1.7 million deaths reported which is on the rise [2]. One of the main ways to prevent this disease is social distancing which has led to fundamental changes in all aspects of our lives [3]. One aspect that has been markedly unstable is traditional educational practices. The time course of these changes is indeterminate [4]. These have affected conventional, in-person academic education and training. Hence, there is a persistent need to innovate and implement alternative educational and assessment strategies [5]. Many governments have ordered educational centers to cease face-to-face instruction for most of their students, requiring them to switch to online teaching and virtual education [6]. In Iran, due to the prevalence of COVID-19 and its high mortality rate (about 450 deaths per day), distance education is the preferred education policy during the quarantine period [7]. A variety of electronic methods and strategies can be used to continue academic education during the pandemic. Methods such as Virtual classrooms, flipped classrooms, Unified communication, and collaboration platforms like Microsoft Teams, Google Classroom, and Blackboard. They contain options of office chat, video conferencing, and file storage spaces that remain course structured and are easy to work with. They usually support uploading and sharing a diversity of contents including Word, PDF, Excel files, audio, videos, etc. These also let the tracking of undergraduate learning and evaluation by means of quizzes and also the rubric-based evaluation of submitted coursework [8].

Videos are extensively intended for supporting and stimulating student perception in a variety of conditions, especially in distance education [9]. Various studies have shown that video training is a powerful instrument for education and acquisition of clinical skills [10–12]. Other studies indicate that, for improving the effectiveness of educational videos, two main elements must prevail: adhering to the content and ensuring maximum interactivity [13]. Previous studies showed that a video-based education activity was useful for physiotherapy students in preparation for practical examinations [14, 15].

As part of the specialized education process for students to emphasize the importance of apprenticeship in medical education, and the value of work-based learning, some universities offer distance or virtual apprenticeship during the Covid-19 epidemic. Unlike many undergraduate courses, turning an apprenticeship into an online method is relatively easy. In fact, many apprenticeships take place online and include periodic zoom or Skype sessions between faculty members and students, a specific program for preparing reports and reflections at the end of the apprenticeship, and an uploading system [16–18].

The apprenticeship stage is also included in the educational programs of Iranian universities as activities that have facilitated performing skills in a real environment. In the course of Public Health, immunization, and vaccination skills, one of the main tasks of the graduates of this course, training was provided in a practical way in interaction with the environment, instructors, and staff of health care units [19]. However, at present, with the outbreak of Covid-19 in Iran, all theory courses are available online, and practical skills training have been suspended.

Practical immunization training for Iranian public health students includes teaching the types of vaccines, the nature of vaccines, the cold chain, vaccine injection, Safe vaccine handling, the injection site, vaccination schedule, and common side effects of vaccination.

According to the guidelines, vaccination has a more significant impact on the prevention of infectious diseases if it is carried out in accordance with international and national policies. Therefore, professional knowledge and skills through appropriate training are crucial for public health students who are responsible for all age groups [20].

Since practical training is an essential component of implementing expanded immunization programs in quarantine conditions in Iran, the aim of this study is to determine the effectiveness of online educational videos as an instrument of distance education.

## Materials and method

### Ethics approval and consent to participate

The Ethics Committee of Hamadan University of Medical Sciences approved this study (IR. UMSHA.REC.1394.588). The participants signed a written informed consent in which they were explained the study objectives, the risks and benefits, and the voluntary nature of participation in the study. Data were collected from the study participants. The students in the control group were provided with the video after completing the post-test. The trial was registered under the following code: IRCT2016110427488N1 (12/04/2020).

### Design of the study

Two-group design was used in which a two-arm randomized trial was selected and performed in June 2020 (The academic year in Iran begins in the mid-half of September every year and continues until the mid-half of August of the following year) at the Undergraduate Public Health Course at Hamadan University of Medical Science. Due to the Covid-19 outbreak and lack of in-person apprenticeship, it was not possible to select a control group from the current semester students that passed vaccination skill training in the Health Centers. For this reason, the control group was selected from students of the same course who had passed the in-person vaccination apprenticeship in the previous year.

This randomized controlled clinical trial was performed on 120 Undergraduate Public Health students at Hamadan University of Medical Science in Hamadan in the west of Iran during January 2020.

### Participants

During the internship period, students attend comprehensive health centers and health homes and provide the necessary services to clients exactly like other healthcare workers and health staff in health centers under the supervision of a trainer.

In the Health Faculty of Hamadan University of Medical Sciences, before starting the internship, students spend the related chapters in face-to-face workshops to familiarize themselves with the programs implemented in the country's health system.

These headings include programs to combat diseases (communicable and non-communicable), children's health program, mother's health program, vaccination program, middle-aged health program-elderly program, and adolescent program-youth program.

In this study, the students of two classes (a discontinuous Bachelor class (67 people)—a Continuous Bachelor class (63 people)) of Hamadan University of Medical Sciences Faculty of Health, who were in the 7th semester and had an internship unit in the field, were selected for the study. Three students were excluded from the study because they were employees of the health system and four students were excluded because they had completed their Compulsory medical service program.

The educational programs of these two classes were held separately. For this purpose, these two classes were randomly allocated into intervention and control groups. The intervention group received the vaccination training program through video and online. The control group received the same routine workshop program. Three persons from the intervention group were excluded from the study due to the incomplete pre-test and post-test questionnaires.

Fig 1 presents a flowchart of class' and student recruitment, student allocation into 2 studies groups.

The sample size was calculated according to the following equation: Sample size = $(z_{1-\alpha/2}+z_{1-\beta})^2 (\delta_1^2+\delta_2^2)/ (\mu_1- \mu_2)^2$.

The standard deviations were considered to be $\delta_1 = 1.37$ and $\delta_1 = 1.18$ based on the previous studies [21]. The difference of means was 0.72, and the confidence interval was considered at 95%. Non-response error of 10% was included. In this research, 60 students were included in each group.

Inclusion criteria were: consent to participate in the study, and completion of Internship 1.

Exclusion criteria were: completion of the compulsory medical service program, and employment in the health system.

The students in the intervention and control groups were evaluated in terms of Knowledge, self-efficacy and attitude before and immediately after the intervention. Their performance in the vaccination was also measured a month after the intervention was completed and before they started vaccination work in comprehensive health centers or health homes. The performance was determined by a checklist, which was completed by one of the researchers in both groups.

## The online material

The intervention included teaching the immunization program through an online educational video based on Bandura's theory of self-efficacy. This educational video was prepared by the research group of Health faculty based on the Immunization handbook of the Iranian Ministry of Health and Medical Education and World Health Organization (WHO) instructions, which included two general sections:

A) Importance of vaccination, familiarity with cold chain equipment, familiarity with cold chain monitoring tools, knowledge of different types of vaccines, their nature, and location in the refrigerator.

B) Practical training of vaccination interactively based on the theory of self-efficacy, including all the necessary skills for injecting different types of vaccines, including familiarity with how to inject different types of vaccines, familiarity with the appropriate time, dose, place, and angle of each vaccine and how to communicate properly with children at the time of vaccination.

All the processes were performed gradually with the help of a health center staff. All the steps were filmed. In order to make this educational video, the relevant scenario was first prepared by the research group, and after its scientific approval by the professors of the Faculty of Health, it was directed and the video was filmed using a professional team.

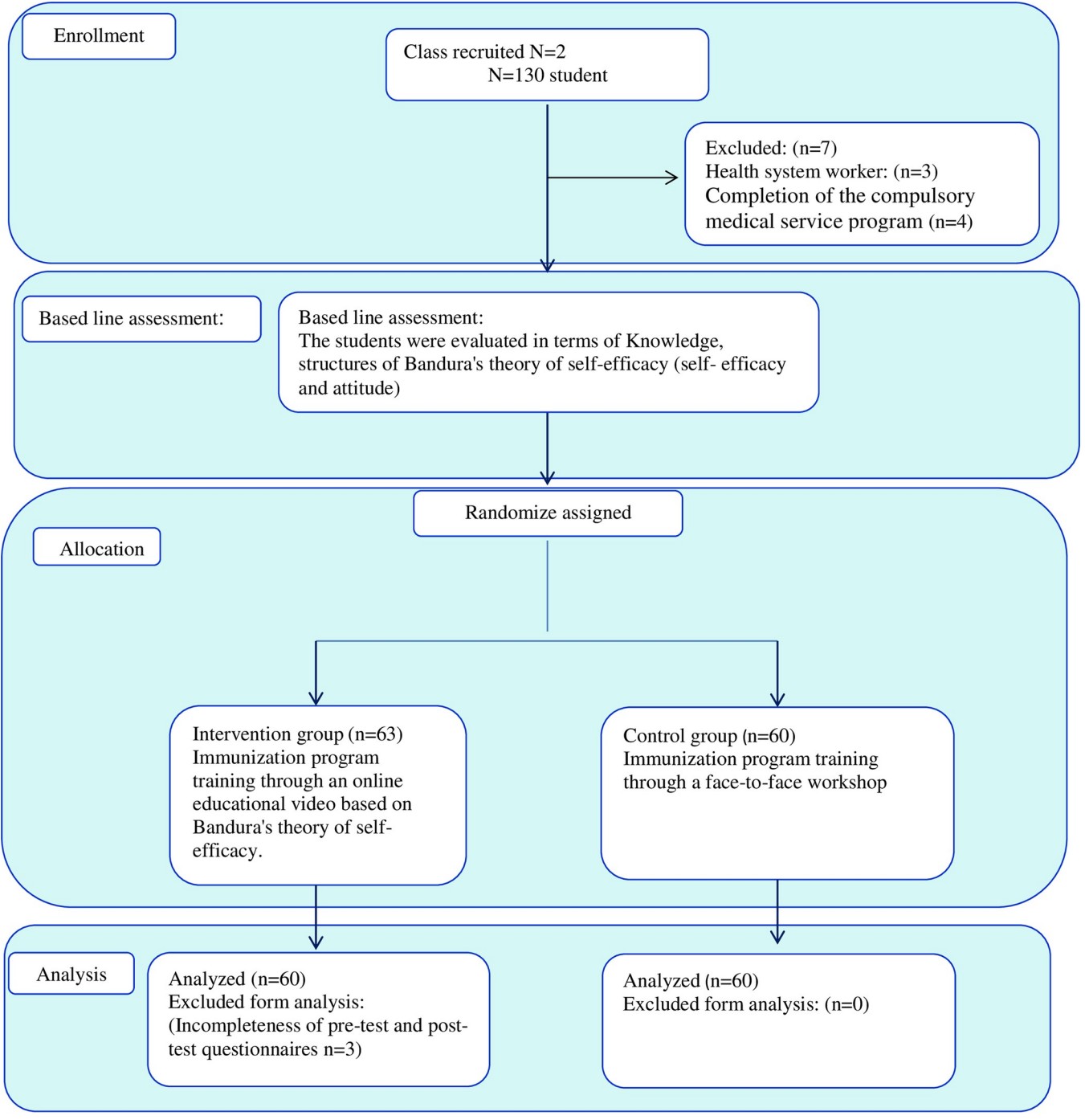

**Fig 1. CONSORT flow diagram of participants throughout the study.**

## Intervention

The educational video was shown to intervention group students via Navid (an online education site of medical science universities in Iran, which was launched during the Cocid-19 pandemic), two sessions a week, for two weeks. The duration of each session was two hours.

The video was prepared interactively in which, after the training, the students were asked to practice the educated skills themselves, then the video was stopped by the instructor, and each student carried out the requested activity online. Their feedback helped the instructors to resolve their problems. The training videos were 30 minutes long. For better management of sessions, the number of participants in each online session was 15 (each session was repeated 4 times). All the students were advised to prepare syringes and vials and use dolls to practice injecting vaccination at home.

The students of the control group had trained in immunization and vaccination in person at health centers, in the previous semester.

## Data collection instruments

All data collection steps were performed online. The data collection instrument included a three-part questionnaire and a performance evaluation checklist. All data collection steps were performed online.

To send and receive the questionnaires online, an application designed at Hamadan University of Medical Sciences for this purpose was used.

To complete the checklist online, Skype software was used, in which each student was asked to perform vaccination practically and the checklist items were scored based on the student's performance.

## A) Questionnaire

The questionnaire consisted of three parts:

The first part of the questionnaire: includes 21 questions related to knowledge of immunization, which for each correct answer was assigned a score of 1 and otherwise a score of 0. The maximum score for this section was 21.

The second part of the questionnaire: includes 6 questions related to the attitude towards Immunization, which is designed based on the 5 Likert scales (strongly agree, agree, have no opinion, disagree, and strongly disagree). The maximum score was 30 and the minimum was 6.

The third part of the questionnaire: Self-efficacy questions related to the vaccination ability that includes 9 questions based on the 5 Likert scales (strongly agree, agree, have no opinion, disagree, and strongly disagree). Scores ranged from 9 to 45.

The content validity method was used to test the validity of the questionnaire. Thus, the questionnaire was prepared based on the instructions of the Iran Ministry of Health then its content was inspected by 2 faculty members of the School of Health and the necessary corrections were considered. To evaluate the reliability of the questionnaire, so the questionnaire was completed by 30 students and Cronbach's alpha was determined as 0.76 for knowledge, 0.72 for attitude and 0.76 for self-efficacy, and 0.75 for the whole questionnaire. Questionnaires were designed and completed online using Google Docs.

## B) Performance evaluation checklist

A performance checklist containing 24 questions was designed to evaluate students' skills in vaccination, which was completed 1 month after the intervention by one of the members of the research team in the form of observation for students both in the control and intervention group; The checklist for each question has 3 options "completely, somewhat, and at all", if the relevant item is completed by the student, the option was "completely", if half of the relevant item was done, "somewhat" and if the relevant item is not done, the option was " at all". It was

necessary to explain that by using Skype software, the student's performance on the Marquette or doll was measured using real needles and vials.

The face validity of the performance checklist was confirmed by the research professors. Reliability and validity of the questionnaires were performed by test-retest reliability and internal consistency. For this purpose, the internal correlation coefficient and Cronbach's alpha tests were used, respectively. In the retest method in 2 stages 10 days apart, questionnaires were completed by 20 students with similar conditions. Retest coefficients between the obtained scores were calculated using ICC then values of 0.6 or higher were accepted [22]. Data were analyzed using SPSS-16 software. Paired t-test, independent t-test, and Chi-square. All methods were carried out in accordance with relevant University guidelines and regulations.

## Results

All students in the sample were fourth-year public health students. Table 1 summarizes the demographics for all students. There were no statistically significant differences between the intervention and control groups with respect to the demographic factors associated with knowledge, attitude, self-efficacy, and practice (Table 1).

There were no statistically significant differences in knowledge (P = 0.65), attitude (P = 0.39), and self-efficacy (P = 0.82) between the intervention and control groups before the intervention (Table 2). According to the results, there were statistically significant differences between the intervention and control groups regarding knowledge (P<0.001), attitude (P = 0.01), and self-efficacy (P<0.001) after the intervention (Table 2). The mean score of knowledge and self-efficacy in both groups increased statistically significantly after the intervention. The increase in knowledge in the intervention group was 43.1% and in the control group was 26.8%. In addition, the increase in self-efficacy score was 30.8% in the intervention group and 9% in the control group (Table 2).

The results of this study showed that there was a statistically significant difference between the intervention and control groups after the educational intervention in the mean performance score (P <0.001, Table 3).

## Discussion

The COVID-19 pandemic is a huge challenge to education systems. Most governments played catch-up to the exponential spread of COVID-19, so institutions had very little time to prepare for a remote-teaching regime, especially in practical courses [5].

**Table 1. Demographic characteristics of participants.**

| Variable | | Intervention | Control | P.Value[a] |
|---|---|---|---|---|
| | | Mean (SD) | Mean (SD) | |
| **Age**(year) | | 22.41(0.49) | 22.36(0.48) | 0.57 |
| **Grade point average** (GPA) | | 15.87(0.70) | 14.69(1.70) | 0.15 |
| | | Intervention | Control | P.Value |
| | | N (%) | N (%) | |
| **Sex** | Male | 12(20) | 15(25) | 0.51 |
| | Female | 48(80) | 45(75) | |

[a]. Independent Samples T Test

[b]. Chi-square

**Table 2. Distribution of knowledge, attitude, and self-efficacy before and after intervention in the two groups.**

| Variable | Group | Control | Intervention | P.Value [a] | Re-range Scores[c](%) | | |
|---|---|---|---|---|---|---|---|
| | | Mean (SD) | Mean (SD) | | Control | Intervention | Difference |
| knowledge | Before | 10.41(2.32) | 10.12(2.43) | 0.65 | 49.5 | 48.1 | 1.4 |
| | After | 16.03(3.00) | 19.17(0.92) | < 0.001 | 76.3 | 91.2 | 14.9 |
| | Difference | 5.62 | 9.05 | < 0.001 | 26.8 | 43.1 | |
| | P.Value [b] | < 0.001 | < 0.001 | | | | |
| Attitude | Before | 21.97(2.70) | 22.56(2.34) | 0.39 | 66.5 | 69 | 2.5 |
| | After | 20.28(3.25) | 22.56(2.95) | 0.01 | 59.5 | 69 | 9.5 |
| | Difference | -1.69 | 0 | 0.01 | -7 | 0 | |
| | P.Value | 0.057 | 0.99 | | | | |
| Self-Efficacy | Before | 30.21(7.61) | 29.76(7.32) | 0.82 | 58.9 | 57.6 | 1.3 |
| | After | 33.45(4.83) | 40.84(3.71) | < 0.001 | 67.9 | 88.4 | 20.5 |
| | Difference | 3.24 | 11.08 | < 0.001 | 9 | 30.8 | |
| | P.Value | < 0.001 | < 0.001 | | | | |

[a]. Independent Samples T Test

[b]. Paired sample T test

[c]. The scores between two groups, I.e., intervention and control groups, re-change to 0–100 for analysis

The aim of this study was to determine the effect of using online educational videos on the learning of vaccination practical skills in apprenticeship courses for public health undergraduate students during the coronavirus epidemic.

In general, results from the present study provide objective evidence that online video-based education can be as effective as standard teaching methods, which is consistent with the current literature [23, 24]. Instructional videos can be an effective complement to current methods, freeing up class time for more engaging and interactive topics. They can be turned into a cost-effective teaching method so that video-based libraries can be created and used by a large number of the academic educators [25]. In the field of medical education, the development of online simulators in the field of medicine, the promotion of virtual hospitals and telemedicine, providing virtual cases, and holding online exams can help promote virtual education [26]. In contrast, in the study of Chakraborty et al. Students stated that online education was stressful for them and had a negative impact on their health and social life [27]. In another study, the cause of students' stress and anxiety during online learning in the Covid-19 lockdown period, lack of equal access to digital technologies, as well the lack of skill in using online education facilities, is stated [28].

In the present study, there was a statistically significant difference between the two groups so the mean performance score was higher in the intervention group. As noted in other studies, and specifically described by Bennett [29], students reported that using video as a learning tool seemed more effective because they could easily see what was in the film. Descriptive studies and various expert opinions state that online learning is an important opportunity for

**Table 3. Comparison of the mean score of students' performance after the intervention in the two groups.**

| Variable | Control | Intervention | P.Value[a] |
|---|---|---|---|
| | Mean (SD) | Mean (SD) | |
| Performance | 22.38(5.40) | 27.92(6.00) | < 0.001 |

[a]. Independent Samples T Test

students to enhance and support learning, so it helps to make this method work better. However, this method cannot completely replace face-to-face courses, although it widely helps and complements the training. Online education certainly provides significant benefits for learning, even with limitations such as the need for high-speed Internet and the provision of peripherals for students (Smart Phone, tablets, or laptops) [30]. This result contrasts previous findings showing mostly detrimental effects of school closures on students' performance and well-being [13, 31]. For example, Azevedo al. showed that performance on national exams in the Netherlands decreased after the shutdown [32]. However, this difference is justified by the fact that student's ages, educational contexts, and subjects in these studies are different.

Knowledge and attitude mean scores were enhanced in the present study which is in line with other studies [33]. This finding can be due to the possibility of repeatedly downloading educational videos, uploading additional content on the university educational website, and the possibility of viewing the educational video at the best time according to students [34, 35]. Additionally, in a systematic review, it was revealed that online e-Learning does lead to positive changes in knowledge, attitude, and satisfaction and seems to be more effective than traditional learning in terms of knowledge gained [36]. This finding is not consistent with Etajuri et al. in which only 49.7% of the dentistry students were satisfied with the clinical knowledge delivered through online classes. This fact may be related to the limited manual training and lack of interaction with the patients [37].

The result of the present study revealed that there was a significant difference in self-efficacy between the two groups, which can be attributed to details such as the use of verbal encouragement for students during the performance, dividing activities into smaller steps, and students practicing in their own homes and away from the stressful academic environment [38]. This finding was not consistent with Moeini et al. in which web-based programs were used to improve the self-efficacy of adolescent girls to deal with depression [39], and this inconsistency can be due to the nature of the intervention outcome because studies have shown that a long duration of the intervention is necessary for improving self-efficacy in depression [40, 41].

The findings of this study showed that online e-learning programs can be useful in training healthcare professionals without successive investment. The findings of the present study showed that e-learning is as effective as traditional learning and has many advantages compared to traditional learning. Universities can adopt these technologies and can reach a wider audience within and outside their country, thus offering a tremendous growth opportunity for educational institutions.

Limitations of the current study included: 1- Problems related to conducting practical training in person for the control group due to the differences between students in the educational semester and the outbreak of COVID-19. 2- Problems related to online education implementation infrastructure. 3- Possibility of unnoticed variables that have not been measured in this study. In this regard, cohort intervention studies could be considered. In addition to these, it is recommended that future studies measure the student's satisfaction with the use of online learning methods with appropriate tools, and also examine the effect of the participants' skills in using new technologies on the level of learning.

## Conclusion

Although the Covid-19 pandemic has posed many problems in all aspects of society, including public health, it has proposed some educational competencies, including pervasive access to learning. It seems that virtual education has entered a new phase and more attention is attached to distance learning activities. On the other hand, policymakers have produced a large body of policy work on virtual learning. Therefore, we are expected to see its prosperity

by developing the required infrastructure, including the development of high-speed internet networks, the production of interactive learning software, and the use of pandemic experiences.

## Supporting information

**S1 File. Demographic variable, knowledge, attitude and self-efficacy control & intervention group.**
(SAV)

**S2 File. Practice control & intervention group.**
(SAV)

## Acknowledgments

The authors would like to thank all students who helped in distributing and collecting the data.

## Author Contributions

**Conceptualization:** Samane Shirahmadi, Seyed Mohamad Mehdi Hazavehei.

**Data curation:** Tahere Etesamifard, Neda Dadaei, Malihe Taheri.

**Formal analysis:** Hamid Abbasi, Ghodratolah Roshanaei.

**Funding acquisition:** Seyed Mohamad Mehdi Hazavehei.

**Investigation:** Samane Shirahmadi, Hamid Abbasi, Malihe Taheri.

**Methodology:** Samane Shirahmadi, Ghodratolah Roshanaei, Malihe Taheri.

**Resources:** Marzie Otogara, Neda Dadaei, Malihe Taheri.

**Software:** Marzie Otogara, Tahere Etesamifard.

**Supervision:** Seyed Mohamad Mehdi Hazavehei.

**Validation:** Tahere Etesamifard, Ghodratolah Roshanaei.

**Writing – original draft:** Samane Shirahmadi, Marzie Otogara, Malihe Taheri.

**Writing – review & editing:** Samane Shirahmadi, Malihe Taheri.

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
