## [Decision Letter · Decision Letter 0]

23 Jun 2022

Dear Dr. Malihe Taheri,

Thank you for submitting your manuscript to PLOS ONE. After careful consideration, we feel that it has merit but does not fully meet PLOS ONE’s publication criteria as it currently stands. Therefore, we invite you to submit a revised version of the manuscript that addresses the points raised during the review process.

We look forward to receiving your revised manuscript.

Kind regards,

Prabhat Mittal, Ph.D.

Academic Editor

PLOS ONE

Journal Requirements:

2. Please provide additional details regarding participant consent. In the ethics statement in the Methods and online submission information, please ensure that you have specified what type you obtained (for instance, written or verbal, and if verbal, how it was documented and witnessed). If your study included minors, state whether you obtained consent from parents or guardians. If the need for consent was waived by the ethics committee, please include this information

Additional Editor Comments (if provided):

You can include the following the reference to your manuscript

Yadav, S., Chakraborty, P., Meena, L., Yadav, D., & Mittal, P. (2021). Children’s interaction with touchscreen devices: Performance and validity of Fitts’ law. Human Behavior and Emerging Technologies, 3(5), 1132–1140. https://doi.org/10.1002/hbe2.305

Yadav, S., Chakraborty, P., & Mittal, P. (2021). Designing Drawing Apps for Children: Artistic and Technological Factors. International Journal of Human-Computer Interaction, 1–15. https://doi.org/10.1080/10447318.2021.1926113

Chakraborty, P., Mittal, P., Gupta, M. S., Yadav, S., & Arora, A. (2021). Opinion of students on online education during the COVID-19 pandemic. Human Behavior and Emerging Technologies, 3(3), 357–365. https://doi.org/10.1002/hbe2.240

Bhatia, A., & Mittal, P. (2019). Big Data Driven Healthcare Supply Chain: Understanding Potentials and Capabilities. SSRN Electronic Journal. https://doi.org/10.2139/ssrn.3464217

Yadav, S., Chakraborty, P., Mittal, P., & Arora, U. (2018). Children aged 6–24 months like to watch YouTube videos but could not learn anything from them. Acta Paediatrica, International Journal of Paediatrics, 107(8), 1461–1466. https://doi.org/10.1111/apa.14291

Verma, C. P., Bansal, R., & Mittal, P. (2020). Control of COVID-19: A Counter Factual Analysis. Administrative Development, Journal of HIPA, Shimla, 7(1), 1–24.

Reviewers' comments:

Reviewer's Responses to Questions

**Comments to the Author**

1. Is the manuscript technically sound, and do the data support the conclusions?

Reviewer #1: Partly

Reviewer #2: Yes

2. Has the statistical analysis been performed appropriately and rigorously? 

Reviewer #1: Yes

Reviewer #2: Yes

3. Have the authors made all data underlying the findings in their manuscript fully available?

Reviewer #1: No

Reviewer #2: Yes

4. Is the manuscript presented in an intelligible fashion and written in standard English?

Reviewer #1: No

Reviewer #2: Yes

5. Review Comments to the Author

Reviewer #1: Manuscript deals with an important issue and has used appropriate methodology and data collection technique. However, some of the areas where correction is needed are listed below -

1. Lot of Grammatical problems are there and whole text needs editing.

2. Apart from T-test, some advanced statistical tests can be applied to make it more technically sound.

3. Authors are advised to support data with existing literature.

Reviewer #2: This is an interesting study and the author have collected a unique dataset using cutting edge methodology. However, I recommend that the conclusion and discussion, as well as the recommendation based on the observations, could be more detailed.

6. PLOS authors have the option to publish the peer review history of their article (what does this mean?). If published, this will include your full peer review and any attached files.

Reviewer #1: **Yes: **Dr Syeedun Nisa

Reviewer #2: No

---

## [Author Response · Author response to Decision Letter 0]

21 Aug 2022

Author’s response to reviews

Title: Effectiveness of online practical education on vaccination training in the students of bachelor programs during the Covid-19 pandemic

Date: 18 August 2022

We thank all the Reviewers for their valuable feedback and taking the time to provide useful comments to improve our manuscript entitled “Effectiveness of Online Practical Education on Vaccination Learning in the apprenticeship stage of Bachelor students in the Covid-19 Pandemic”. Based on the constructive comments the following changes have been made:

It is necessary to explain that the corrections considered by honorable reviewers specified with the yellow highlight in the text of the manuscript. 

Journal Requirements:

1- Please ensure that your manuscript meets PLOS ONE's style requirements, including those for file naming.

Response: Thanks for the valuable comments, the requested corrections were made

2- Please provide additional details regarding participant consent.

Response: The additional details regarding participant consent were added in Method section, page5, line 101-108

3- We suggest you thoroughly copyedit your manuscript for language usage, spelling, and grammar.

 Response: Thanks for the valuable comments of the esteemed Editor that helped to improve the details of the study. The language of manuscript was revised. The track change file of language editing Institute attached as “supporting information” file.

4- In your Data Availability statement, you have not specified where the minimal data set underlying the results described in your manuscript can be found.

Response: Minimal data set of present study uploaded as “Supporting Information” files 

Response: The grant information statement was removed from the manuscript and modified in the submission system. Due to the fluctuation of the price of the dollar against the Rial (Iranian currency), the amount that was previously registered in the system was modified according to the current price of the dollar in Iran.

Response to Reviewer 1: Dr Syeedun Nisa

1-Lot of Grammatical problems is there and whole text needs editing.

Response: Thanks for the valuable comments of the reviewers that helped to improve the details of the study. The language of manuscript was revised. The track change file of language editing Institute attached.

2-Apart from T-test, some advanced statistical tests can be applied to make it more technically sound. 

Response: Thank you for your valuable comment. We examined all variables based on literature that may have an impact on students' knowledge, attitude, self-efficacy and performance.

The results showed that there is no statistically significant difference between the intervention and control groups in terms of these variables before the study.

Therefore, it can be concluded that the difference in the mean scores of knowledge, attitude, self-efficacy and performance between the intervention and control groups is due to the intervention.

We added this sentence in the limitations.

It is possible that there have been variables that have not been measured in this study. This issue is one of the characteristics of cohort intervention studies.

Also, we reviewed many experimental studies (1-10), and independent t-test was used in all of these studies. A sample of studies is given below.

1. Theodosi S, Nicolaidou I. Affecting young children’s knowledge, attitudes, and behaviors for ultraviolet radiation protection through the internet of things: a quasi-experimental study. Computers. 2021;10(11):137.

2. Öz GÖ, Ordu Y. The effects of web based education and Kahoot usage in evaluation of the knowledge and skills regarding intramuscular injection among nursing students. Nurse Education Today. 2021;103:104910.

3. Craig SJ, Kastello JC, Cieslowski BJ, Rovnyak V. Simulation strategies to increase nursing student clinical competence in safe medication administration practices: A quasi-experimental study. Nurse Education Today. 2021;96:104605.

4. Grønlien HK, Christoffersen TE, Ringstad Ø, Andreassen M, Lugo RG. A blended learning teaching strategy strengthens the nursing students’ performance and self-reported learning outcome achievement in an anatomy, physiology and biochemistry course–A quasi-experimental study. Nurse Education in Practice. 2021;52:103046.

5. Chang H-Y, Wu H-F, Chang Y-C, Tseng Y-S, Wang Y-C. The effects of a virtual simulation-based, mobile technology application on nursing students’ learning achievement and cognitive load: Randomized controlled trial. International Journal of Nursing Studies. 2021;120:103948.

6. Putra A, Sumarmi S, Sahrina A, Fajrilia A, Islam M, Yembuu B. Effect of Mobile-Augmented Reality (MAR) in digital encyclopedia on the complex problem solving and attitudes of undergraduate student. International Journal of Emerging Technologies in Learning (IJET). 2021;16(7):119-34.

7. Ma X, Yang Y, Chow KM, Zang Y. Chinese adolescents’ sexual and reproductive health education: A quasi‐experimental study. Public Health Nursing. 2022;39(1):116-25.

8. Sarker R, Islam M, Moonajilin M, Rahman M, Gesesew HA, Ward PR. Effectiveness of educational intervention on breast cancer knowledge and breast self-examination among female university students in Bangladesh: a pre-post quasi-experimental study. BMC cancer. 2022;22(1):1-7.

9. Kandula UR, Philip D, Mathew S, Subin A, Godphy A, Alex N, et al. Efficacy of video educational program on interception of urinary tract infection and neurological stress among teenage girls: An uncontrolled experimental study. Neuroscience Informatics. 2022;2(3):100026.

10. Permatasari TAE, Rizqiya F, Kusumaningati W, Suryaalamsah II, Hermiwahyoeni Z. The effect of nutrition and reproductive health education of pregnant women in Indonesia using quasi experimental study. BMC Pregnancy and Childbirth. 2021;21(1):1-15.

 If the esteemed reviewer has a specific statistical analysis in mind say its name. We will do.

3. Authors are advised to support data with existing literature.

Response: The results of this study were compared with similar studies in the discussion section.

Response to Reviewer 2:

1-This is an interesting study and the author have collected a unique dataset using cutting edge methodology. However, I recommend that the conclusion and discussion, as well as the recommendation based on the observations, could be more detailed.

Response: Thanks for the valuable comments of the esteemed reviewer that helped to improve the study; more details were provided in section of conclusion and discussion, as well as the recommendation based on the observations.

---

## [Decision Letter · Decision Letter 1]

24 Oct 2022

PONE-D-21-36844R1Effectiveness of online practical education on vaccination training in the students of bachelor programs during the Covid-19 pandemicPLOS ONE

Dear Dr. Taheri,

Thank you for submitting your manuscript to PLOS ONE. After careful consideration, we feel that it has merit but does not fully meet PLOS ONE’s publication criteria as it currently stands. Therefore, we invite you to submit a revised version of the manuscript that addresses the points raised during the review process.

We look forward to receiving your revised manuscript.

Kind regards,

Wenping Gong, Ph.D.

Academic Editor

PLOS ONE

Journal Requirements:

Reviewers' comments:

Reviewer's Responses to Questions

**Comments to the Author**

1. If the authors have adequately addressed your comments raised in a previous round of review and you feel that this manuscript is now acceptable for publication, you may indicate that here to bypass the “Comments to the Author” section, enter your conflict of interest statement in the “Confidential to Editor” section, and submit your "Accept" recommendation.

Reviewer #3: (No Response)

Reviewer #4: (No Response)

2. Is the manuscript technically sound, and do the data support the conclusions?

Reviewer #3: (No Response)

Reviewer #4: Partly

3. Has the statistical analysis been performed appropriately and rigorously? 

Reviewer #3: (No Response)

Reviewer #4: Yes

4. Have the authors made all data underlying the findings in their manuscript fully available?

Reviewer #3: (No Response)

Reviewer #4: Yes

5. Is the manuscript presented in an intelligible fashion and written in standard English?

Reviewer #3: (No Response)

Reviewer #4: No

6. Review Comments to the Author

Reviewer #3: This manuscript could be an important reference for future studies. However, minor is still needed to improve the quality of this paper. Please revise the manuscript to address the expressed concerns. After thorough review, I am recommending some revisions. In this regard, kindly address the following comments and suggestions to further improve your manuscript

1. It is better that in abstract result mentioned numerical result .for example prevalence of group1 vs group 2 with p value=?

2. The methods need to be improved by providing more detail information related to participant’s selection (e.g. respond rate; necessary permissions from who? How did the researcher contact the potential participants?)

3. Discuss more about your sampling strategy? The structure of your sampling is so vague and understandable. Did you have sampling frame? how did you access to this frame

4. It was better if you could show the process of samples selection and methods using a flowchart with consort format.

5. What are the data extract’s center characteristics? is it governmental or private, is it referral or not referral and so on, discuss more about it

6. How many observers did you have? if you had more than one observer, you must mention agreement index like kappa coefficient (write in method section)

7. write about all applied exclusion and inclusion criteria a bit more clearly by which you selected samples for this survey.

Reviewer #4: In the manuscript, Taheri and co-workers investigated the effectiveness of practical vaccination education via video training during the COVID-19 pandemic. It is a specific and interesting work and I have one question about the selection of the control group.

The control group was selected from students in the previous year because there was no one passing the in-person training this year. But students can accumulate practical experience or even forget about the knowledge in the class after one year, which may affect the results of knowledge, attitude, and self-efficacy. Is this the best choice for the control group to reflect the video training effectiveness?

Also, some grammar errors in the main text need to be corrected, e.g. line 86-88, line 247. Odd and inconsistent capital letters, e.g. in Table 2 column, line 91, 243, 257, 266 and so on, need to be corrected, too. Please carefully proofread the manuscript.

7. PLOS authors have the option to publish the peer review history of their article (what does this mean?). If published, this will include your full peer review and any attached files.

Reviewer #3: **Yes: **Hadi Tehrani

Reviewer #4: No

---

## [Author Response · Author response to Decision Letter 1]

12 Dec 2022

Author’s response to reviews

Title: Effectiveness of online practical education on vaccination training in the students of bachelor programs during the Covid-19 pandemic

Authors:

Samane Shirahmadi, 

Seyed Mohamad Mehdi Hazavehei

Hamid Abbasi, 

Marzie Otogara, 

Tahere Etesamifard, 

Ghodratolah Roshanaei, 

Neda Dadaei

Malihe Taheri* 

Version: 2

Date: 2022 Des 10

Author's response to reviews: see over

We thank all the Reviewers for their valuable feedback and taking the time to provide useful comments to improve our manuscript entitled “Effectiveness of online practical education on vaccination training in the students of bachelor programs during the Covid-19 pandemic.” Based on the constructive comments the following changes have been made.

Response to Reviewer 3:

1. It is better that in abstract result mentioned numerical result .for example prevalence of group1 vs group 2 with p value=?

Response: Thank you for your valuable comment. We have revised the abstract result section.

2. The methods need to be improved by providing more detail information related to participant’s selection (e.g. respond rate; necessary permissions from who? How did the researcher contact the potential participants?)

Response: Thank you for your valuable comment. We have revised the method section. Page…, line….. 

3. Discuss more about your sampling strategy? The structure of your sampling is so vague and understandable. Did you have sampling frame? How did you access to this frame

Response: Thank you for your valuable comment. We have revised the method section and sampling strategy. 

We have added these paragraphs in in method section:" 

Page7.

4. It was better if you could show the process of samples selection and methods using a flowchart with consort format.

Response: Thank you for your valuable comment. We have added flowchart of sampling with consort format (Fig 1). 

5. What are the data extract’s center characteristics? Is it governmental or private, is it referral or not referral and so on, discuss more about it

Response: Thank you for your valuable comment. All the data were extracted from public health undergraduate students of Hamadan University of Medical Sciences. Hamadan University of Medical Sciences is a public university.

6. How many observers did you have? If you had more than one observer, you must mention agreement index like kappa coefficient (write in method section)

Response: Thank you for your valuable comment. There was only one observer. We have added in the method section. Page 8, line 183

7. Write about all applied exclusion and inclusion criteria a bit more clearly by which you selected samples for this survey. 

Response: Thank you for your valuable comment. We have Added exclusion and inclusion criteria in the method section. Page 7-8.

Response to Reviewer 4:

1. The control group was selected from students in the previous year because there was no one passing the in-person training this year. But students can accumulate practical experience or even forget about the knowledge in the class after one year, which may affect the results of knowledge, attitude, and self-efficacy. Is this the best choice for the control group to reflect the video training effectiveness?

Response: Thank you for your valuable comment. We have completed the selection of the control group. Page 8, lines 175-178.

2. Some grammar errors in the main text need to be corrected, e.g. line 86-88, line 247. Odd and inconsistent capital letters e.g. in Table 2 columns, line 91, 243, 257, 266 and so on, need to be corrected, too. Please carefully proofread the manuscript.

Response: Thank you for your valuable comment. We have edited errors.

---

## [Editor Report · Decision Letter 2]

27 Dec 2022

Effectiveness of online practical education on vaccination training in the students of bachelor programs during the Covid-19 pandemic

PONE-D-21-36844R2

Dear Dr. Malihe Taheri,

We’re pleased to inform you that your manuscript has been judged scientifically suitable for publication and will be formally accepted for publication once it meets all outstanding technical requirements.

Kind regards,

Wenping Gong, Ph.D.

Academic Editor

PLOS ONE
---

## [Editor Report · Acceptance letter]

3 Jan 2023

PONE-D-21-36844R2 

Effectiveness of online practical education on vaccination training in the students of bachelor programs during the Covid-19 pandemic 

Dear Dr. Taheri:

I'm pleased to inform you that your manuscript has been deemed suitable for publication in PLOS ONE. Congratulations! Your manuscript is now with our production department. 

Kind regards, 

on behalf of

Dr. Wenping Gong 

Academic Editor

PLOS ONE